# Radiomics for Differentiation of Pediatric Posterior Fossa Tumors: A Meta-Analysis and Systematic Review of the Literature

**DOI:** 10.3390/cancers15245891

**Published:** 2023-12-18

**Authors:** Alexandru Garaba, Francesco Ponzio, Eleonora Agata Grasso, Waleed Brinjikji, Marco Maria Fontanella, Lucio De Maria

**Affiliations:** 1Department of Surgical Specialties, Radiological Sciences and Public Health, University of Brescia, 25121 Brescia, Italy; marco.fontanella@unibs.it (M.M.F.); or lucio.demaria@hcuge.ch (L.D.M.); 2Unit of Neurosurgery, Spedali Civili Hospital, Largo Spedali Civili 1, 25123 Brescia, Italy; 3Interuniversity Department of Regional and Urban Studies and Planning, Politecnico di Torino, 10129 Torino, Italy; fr.ponzio@gmail.com; 4Department of Pediatrics, Children’s Hospital of Philadelphia, Philadelphia, PA 19146, USA; elegrasso601@gmail.com; 5Department of Neurosurgery and Interventional Neuroradiology, Mayo Clinic, Rochester, MN 55905, USA; brinjikji.waleed@mayo.edu; 6Department of Clinical Neuroscience, Geneva University Hospitals (HUG), 1205 Geneva, Switzerland

**Keywords:** radiomics, machine learning, deep learning, pediatric tumors, posterior fossa, systematic review, meta-analysis

## Abstract

**Simple Summary:**

In this study, we reviewed and analyzed various radiomics models used to identify three of the most common pediatric posterior fossa tumors: medulloblastoma, pilocytic astrocytoma, and ependymoma. We wanted to understand how well these models worked overall. By examining a range of studies, we found that the models could effectively distinguish between these tumor types with high accuracy. Certain types of models and specific features in the images seemed to perform better. The findings suggest that these models can be very useful in accurately diagnosing these pediatric brain tumors. This research could lead to even more precise models in the future.

**Abstract:**

Purpose: To better define the overall performance of the current radiomics-based models for the discrimination of pediatric posterior fossa tumors. Methods: A comprehensive literature search of the databases PubMed, Ovid MEDLINE, Ovid EMBASE, Web of Science, and Scopus was designed and conducted by an experienced librarian. We estimated overall sensitivity (SEN) and specificity (SPE). Event rates were pooled across studies using a random-effects meta-analysis, and the χ^2^ test was performed to assess the heterogeneity. Results: Overall SEN and SPE for differentiation between MB, PA, and EP were found to be promising, with SEN values of 93% (95% CI = 0.88–0.96), 83% (95% CI = 0.66–0.93), and 85% (95% CI = 0.71–0.93), and corresponding SPE values of 87% (95% CI = 0.82–0.90), 95% (95% CI = 0.90–0.98) and 90% (95% CI = 0.84–0.94), respectively. For MB, there is a better trend for LR classifiers, while textural features are the most used and the best performing (ACC 96%). As for PA and EP, a synergistic employment of LR and NN classifiers, accompanied by geometrical or morphological features, demonstrated superior performance (ACC 94% and 96%, respectively). Conclusions: The diagnostic performance is high, making radiomics a helpful method to discriminate these tumor types. In the forthcoming years, we expect even more precise models.

## 1. Introduction

Radiomics, a promising tool in medical imaging, employs a non-invasive approach that can aid in the diagnosis, treatment, and prognosis of various diseases [1]. This rapidly expanding discipline involves extracting high-dimensional quantitative data from medical images, subsequently analyzed using artificial intelligence (AI) techniques, such as machine learning (ML) and deep learning (DL), to define features such as tissue heterogeneity, texture, shape, and intensity [2]. These features reveal insights beyond human vision and minimize inter-observer variability in the diagnostic procedure.

In neurosurgical oncology, radiomics proves beneficial for distinguishing between different types of malignancies [3]. Posterior fossa tumors constitute nearly 50% of all pediatric brain tumors.

Medulloblastoma (MB), ependymoma (EP), and pilocytic astrocytoma (PA) are the three most prevalent pediatric posterior fossa tumors [4]. However, these tumors pose a challenge in differentiation due to their similar clinical presentations and radiological features [5]. Given the distinct treatments and prognoses associated with each, accurate diagnosis becomes crucial [6,7,8].

The efficacy of radiomics in distinguishing between these three forms of posterior fossa tumors in pediatric patients remains unclear. We performed a systematic review and a meta-analysis of the current literature to evaluate the performance of radiomics for differentiation between pediatric MB, EP, and PA of the posterior fossa. The findings of this study will shed light on the efficacy of radiomics in the differential diagnosis of pediatric posterior fossa tumors and will provide useful information for the development of future models.

## 2. Materials and Methods

### 2.1. Literature Search

The systematic review was performed according to the Preferred Reporting Items for Systematic Reviews and Meta-Analysis (PRISMA) guidelines [9]. A comprehensive literature search of the databases PubMed, Ovid MEDLINE, Ovid EMBASE, Web of Science, and Scopus was designed and conducted by an experienced librarian with input from the authors. The following research string was used: “radiomics AND ((medulloblastoma OR ependymoma OR pilocytic astrocytoma) OR (posterior fossa))”. The studies were found using the Medical Subject Heading (MeSH) terms and Boolean operators. A search filter was set to show only publications over the designated period. The search was limited to articles published between 2012 and 2022. The first literature search was performed on 5 June 2023, and the search was updated on 28 July 2023. This study was not registered in any public registry such as PROSPERO. 

Two authors (L.D.M. and F.P.) determined the inclusion criteria for the studies in the literature search process. The following inclusion criteria were used: (1) case series including at least 10 patients, (2) studies reporting exclusively histologically proven medulloblastomas, ependymomas, or pilocytic astrocytomas of the posterior fossa, (3) pediatric patients (i.e., aged ≤ 21 years [10]), (4) studies reporting on radiomics for differential diagnosis of posterior fossa tumors, (5) availability of radiomics performance data for differentiation of these tumors. The exclusion criteria were: (1) case reports or review studies, (2) studies reporting on AI-based models other than radiomics, (3) studies on the radiomics differentiation of other tumor types, and (4) studies not reporting performance data of the radiomics model.

The list of identified studies was imported into Endnote X9, and duplicates were removed. The search results were checked by two independent researchers (F.P. and W.B.) with experience according to the inclusion and exclusion criteria. A third reviewer (L.D.M.) resolved all disagreements. Then, eligible articles were subject to full-text screening. Reference lists of identified studies were also reviewed to identify additional relevant studies. 

### 2.2. Data Extraction

For each study, we abstracted the following baseline information: year of publication, total number of patients, distribution of patients per tumor type, and magnetic resonance imaging (MRI) protocol. As for the radiomics models, we collected information about the AI sub-category (i.e., ML or DL), classification algorithms (i.e., logistic regression (LR), neural network (NN), support vector machine (SVM), naïve bayes (NB), k-nearest neighbor (kNN), multilayer perceptron (MLP), random forest (RF), gradient boost (GB), adaptive boosting (AdaBoost), classification tree (CT), single-layer neural network (SLNN), extreme gradient boosting (EGB), elastic net regression (ENR), linear discriminant analysis (LDA), convolutional neural network (CNN), generative adversarial network (GAN), etc.), best-performing classifier, best-performing features (i.e., textural, geometrical or morphological, voxel intensities-based, etc.), best-performing MRI sequences (i.e., apparent diffusion coefficient (ADC), T1-weighted (T1W), contrast-enhanced T1-weighted image (CE-T1W), T2-weighted (T2W), T2-weighted fluid-attenuated inversion recovery (T2-FLAIR), diffusion-weighted imaging (DWI), dynamic susceptibility contrast (DSC)), and the application of cross-validation analysis (i.e., yes or no). Regarding the performance of the models, we extracted data on the resultant sensitivity (SEN) and specificity (SPE), the positive predictive value (PPV) and negative predictive value (NPV), accuracy (ACC), and area under the curve (AUC).

In studies with overlapping patient populations written by the same authors or institution, we only included the largest or most complete dataset. In cases where the outcomes were separated by study cohorts, we abstracted performance outcomes of validation or test cohorts to perform our meta-analysis.

### 2.3. Outcomes

Our primary outcomes were the SEN, SPE, and summary receiver operating characteristics (SROC) curve of radiomics for the differentiation of the pediatric posterior fossa tumors included. Bivariate analyses according to the discrimination task between MB vs. non-MB, PA vs. non-PA, and EP vs. non-EP were conducted. In terms of the performance of the models, we also looked at the PPV, NPV, ACC, and AUC.

The impact of the following variables on the performance of the proposed radiomics models was evaluated as a secondary outcome: year of development, cohort size, AI sub-category, best-performing classifiers, features, and MRI sequences, and application of cross-validation. These variables were also studied quantitatively to picture the current trends of radiomics models differentiating pediatric posterior fossa tumors.

### 2.4. Study Risk of Bias Assessment

We modified the Newcastle–Ottawa Scale (NOS) to assess the methodologic quality of the studies included in our meta-analysis [11]. This tool is designed for use in comparative studies. However, as there was no control group in our studies, we assessed their methodologic quality based on selected items from the scale, focusing on the following questions: (1) Did the study include all patients or consecutive patients vs. a selected sample? (2) Was the study retrospective or prospective? (3) Was clinical follow-up satisfactory, allowing for the ascertainment of all outcomes? (4) Were the outcomes reported? (5) Were there clearly defined inclusion and exclusion criteria [12]?

### 2.5. Statistical Analysis

For the meta-analytic purpose, we considered the total number of patients included in each study’s test or validation dataset. Performance data of the models were figured according to class, i.e., MB vs. non-MB, PA vs. non-PA, and EP vs. non-EP. Data from primary studies were reported in a 2 × 2 contingency table consisting of true positive (TP), false positive (FP), false negative (FN), and true negative (TN) based on the concordance between the biopsy results and radiomics tool predictions. Such a table served as the input for the R-package mada [13], used for modeling the joint estimates of SEN and SPE and their 95% confidence intervals (CIs). Event rates were pooled across studies using a random-effects meta-analysis, and the χ^2^ test was performed to assess the heterogeneity of SEN and SPE, considering the null hypothesis as equality in each case.

To better show the diagnostic performance of AI-based radiomics tools, we made the following further figures of merit: (1) univariate graphics in the form of forest plots for both SEN and SPE; (2) endpoints of interest with individual confidence regions; (3) SROC curve seeking to combine the ROC curves of primary studies. In these last two graphical outcomes, the coordinates of the endpoints of interest are in the form of SEN and 1–SPE, the latter better known as the false positive rate (FPR).

## 3. Results

### 3.1. Literature Review

A total of 268 papers were identified after duplicate removal. After title and abstract analysis, 24 articles were identified for full-text analysis. Eligibility was ascertained for nine articles [14,15,16,17,18,19,20,21,22]. The remaining 15 articles were excluded for the following reasons: (1) studies not reporting data on radiomics performance for the differentiation of posterior fossa tumors (eight articles), (2) studies not reporting on the types of posterior fossa tumors (three articles), (3) studies reporting on AI-based models other than radiomics (three articles), (4) an improper study design (one article). All studies included in the analysis had at least one or more outcome measures available for one or more of the patient groups analyzed. Figure 1 shows the flow chart according to the PRISMA statement [9].

### 3.2. Baseline and Radiomics Data

A total of 1796 patients were included in the systematic review. Most studies were published in 2020 (45%), followed by 2021 and 2022 in equal percentages (22%), and 2015 (11%). The smallest study included 44 patients [16], while the largest had 535 [21]. A total of 911 patients had histology-proved MBs (51%), 531 PAs (30%), and 354 EPs (20%). Differentiation in three posterior fossa tumor types was reported for 1745 patients (97%), of which 884 had medulloblastomas (51%), 330 had ependymomas (19%), and 531 pilocytic astrocytomas (30%). On the other hand, discrimination between two tumor types was reported for 51 patients (3%), of which 27 had medulloblastomas (53%) and 24 had ependymomas (47%). Each study included different MRI sequences, and the most common was T2W (25%), followed by ADC (22%), CE-T1W (19%), and T1W (16%). Other MRI modalities were DWI (9%), T2-FLAIR (6%), and DSC (3%).

A total of eight studies reported on ML (89%), one article on DL (11%), and there were no hybrid studies. As for the classifiers, SVM and RF were the most adopted (19%), followed by kNN (15%), LR (11%), AdaBoost (8%), and others. A summary of the included studies is provided in Table 1.

### 3.3. Primary Outcomes

In total, the performance of radiomics to discriminate by class MB, PA, and EP was reported for 624, 396, and 245 patients, respectively, forming the validation or test datasets of the studies included in our meta-analysis. The overall SEN was 93% (95% CI = 0.88–0.96) for MB, 83% (95% CI = 0.66–0.93) for PA, and 85% (95% CI = 0.71–0.93) for EP. The overall SPE was 87% (95% CI = 0.82–0.90) for MB, 95% (95% CI = 0.90– 0.98) for PA, and 90% (95% CI = 0.84–0.94) for EP. Figure 2 shows the SEN and SPE forest plots of the bivariate analyses for discrimination of MB, PA, and EP. Figure 3 provides the individual confidence regions, and Figure 4 the corresponding SROC curve for the differentiation tasks. Specifically, the summary estimate coordinates of the SROC curve for MB, PA, and EP were [0.92; 0.14], [0.83; 0.05], and [0.83; 0.10], respectively.

For MB, the PPV ranged from 75% to 95%, NPV from 76% to 100%, ACC from 82% to 96%, and AUC from 0.91 to 1.00. For PA, the PPV ranged from 77% to 96%, NPV from 82% to 97%, ACC from 82% to 94%, and AUC from 0.94 to 0.99. Finally, regarding EP, the PPV ranged from 42% to 86%, NPV from 85% to 100%, ACC from 80% to 96%, and AUC from 0.84 to 0.99. Table 2 summarizes the performance data of the radiomics models analyzed. 

### 3.4. Secondary Outcomes

We put into effect a moderator analysis in the form of a diagnostic bivariate meta regression investigating the following factors: year of publications, best-performing classifier, artificial intelligence paradigm (i.e., machine or deep learning), best-performing feature, best-performing image modality and cross-validation (on/off). Our moderator analysis could not evidence any variables to significantly impact the performance of the models. Nonetheless, the best-performing variables for MB (ACC 96%) were the MRI T1W and T2W sequences in combination, the LR classifier, and textural features. Conversely, the best-performing variables for PA (ACC 94%) and EP (ACC 96%) were the MRI T2W sequence alone, the LR and NN classifiers in combinations, and geometrical/morphological features. Most of the studies applied a cross-validation setting (67%), although no study provides any further validation on external datasets. Table 3 summarizes the secondary outcomes.

### 3.5. Study Heterogeneity

The χ^2^ test suggested substantial heterogeneity of SEN and SPEC for the discrimination of MB, PA, and EP.

## 4. Discussion

The systematic review and meta-analysis conducted here provide substantial insights into the promising role of radiomics in effectively differentiating between three prevalent posterior fossa tumors in pediatric patients. We comprehensively identified and analyzed the literature, ultimately yielding a valuable dataset of 1796 patients. The studies, spanning 2015 to 2022, collectively showcased the efficacy of radiomics in distinguishing between MB, PA, and EP. Our analysis revealed that radiomics-based models overall have favorable SEN and SPE in discerning these tumor types, despite variations in specific MRI sequences and classification algorithms. Specifically, the combination of MRI T1W and T2W sequences, the LR classifier, and textural features yielded the best results for MB. Conversely, the MRI T2W sequence alone, a combination of LR and NN classifiers, and geometrical/morphological features demonstrated optimal performance for PA and EP. These outcomes highlight the potential of radiomics for accurate posterior fossa tumor classification in pediatric patients.

### 4.1. Radiomics Models

Our study encompassed a comprehensive analysis of 1796 patients across various years of publication, with a notable concentration of studies emerging in 2020 (45%), closely followed by an equitable distribution between 2021 and 2022 (22% each), and fewer representation in 2015 (11%).

A noteworthy observation is the prevalence of ML techniques (89%) over DL approaches. This could be attributed to the relatively limited availability of large, annotated datasets required for training deep learning models effectively [23]. As deep learning methodologies evolve and more data become available, exploring the potential benefits of deep learning models in this context could provide new avenues for improving tumor classification accuracy.

SVM and RF emerged as the most commonly employed classifiers, each utilized in 19% of the studies. The preference for these classifiers could be attributed to their robustness and ability to handle complex data relationships [24]. Other classifiers like kNN, LR, and AdaBoost were also utilized, each contributing to the diversity of model architectures.

The selection of MRI sequences showcased varying preferences. The T2W sequence was the most common choice, adopted by 25% of the studies, followed by ADC and CE-T1W sequences at 22% and 19%, respectively.

### 4.2. Radiomics Performance

Our findings reveal promising results for radiomics to distinguish between MB, PA, and EP. The overall SEN and SPE of the radiomics models for discriminating MB were 93% and 87%, respectively, while for PA, they were 83% and 95%, and for EP, 85% and 90%, respectively.

Comparing our results with the existing literature, we observed a notable improvement thanks to the discriminative potential of radiomics models. Previous non-radiomics studies reported a lower SEN and SPE when distinguishing between pediatric posterior fossa tumors. For instance, one study achieved an SEN of 81.82% and an SPE of 76.47% for MB, and an SEN of 66.67% and SPE of 89.47% for PA via ADC [25]. Similarly, another study reported a slightly lower SEN (84.6%) and SPE (80%) to distinguish MB and PA, utilizing DWI [26]. The radiomics-based approach demonstrates superior performance, suggesting that the integration of advanced image analysis promises to enhance the accuracy of tumor differentiation.

Our study demonstrated varying performance across different tumor types. The radiomics-based models showed a higher SEN in discriminating MB, potentially due to distinct morphological and textural features associated with this tumor type. In contrast, the differentiation of PA and EP presented somewhat lower SEN values, which could be attributed to their overlapping radiological characteristics and variable tumor heterogeneity [27]. However, despite these challenges, our study highlights that radiomics still offers valuable discriminatory power for these complex tumor differentiations.

The study also employed SROC curves for each tumor type that depicted distinct shapes and trends. Particularly, the SROC curve for PA exhibited a significant shift toward the upper left corner of the plot, indicating a better diagnostic performance compared to MB and EP. This shift suggests its potential to be more effective in correctly identifying TPs while maintaining a reasonable level of TNs. On the other hand, the SROC curves for MB and EP showed a slightly wider spread, with the curves extending more toward higher FPRs or lower SEN values. The SROC curves indicated a potential for future improvement, particularly in the discrimination of PA tumors.

While our findings indicate substantial progress in pediatric posterior fossa tumor differentiation, there are still areas that warrant further investigation. Future studies should focus on refining feature extraction methods, exploring hybrid approaches that integrate radiomics with other advanced imaging techniques, and incorporating larger multicenter datasets to enhance the generalizability of the developed models.

### 4.3. Determinants of Performance

While our subgroup analysis did not unveil statistically significant variables influencing the models’ performance, closer examination of the best-performing features shed light on specific trends contributing to the enhanced ACC.

For the discrimination of MB, the combination of T1W and T2W sequences emerged as a potent predictor, yielding an accuracy of 96%. T1W images are proficient at highlighting anatomical structures and providing information about tissue composition, while T2W images excel at revealing tissue contrast and edema. The association of these sequences could offer a comprehensive portrayal of the tumor’s structural and textural attributes, thereby enhancing the models’ ability to discern subtle differences between tumor types.

Interestingly, the T2W sequence alone yielded the best performance for PA and EP, achieving ACCs of 94% and 96%, respectively. The potential of the T2W sequence might be attributed to its sensitivity in capturing variations in tissue properties, such as cystic and necrotic regions often present in these tumor types [28,29]. The rich textural information conveyed by the T2W sequence could facilitate the identification of unique patterns associated with PA and EP, contributing to their successful differentiation.

Furthermore, the choice of classifiers appeared to play a role in achieving a high ACC. The LR with NN classifiers demonstrated exceptional performance in conjunction with the most informative sequences. LR is known for its simplicity and interpretability, making it well-suited for scenarios where the relationships between features and outcomes can be effectively captured linearly [30]. On the other hand, NN possess the capacity to capture intricate non-linear patterns within data, potentially explaining their efficacy when paired with the best-performing sequences for PA and EP [30].

The most effective features were textural and geometrical/morphological features. Textural features play a crucial role in capturing patterns and variations in image textures. These features provide quantitative information about the spatial arrangement of pixel intensities in an image, reflecting the underlying tissue properties. Common textural features are the Gray-Level Co-occurrence Matrix (GLCM), Gray-Level Run-length Matrix (GLRLM), and Gray-Level Size Zone Matrix (GLSZM) [31]. On the other hand, geometrical/morphological features capture the spatial arrangement and shape characteristics of the regions within an image. These features can provide insights into the overall structure of the observed objects. Typical geometrical/morphological features are compactness, sphericity, and density [31].

Most of the studies adopted cross-validation methodologies (67%), enhancing the reliability and generalizability of the models. Cross-validation minimizes the risk of overfitting, a concern when dealing with relatively small datasets. By partitioning the data into training and validation sets multiple times, cross-validation ensures that the models’ performance metrics are more reflective of their true capabilities when applied to unseen data [32].

### 4.4. Limitations

Despite the number of patients included in our study, this meta-analysis was based on retrospective cohort studies, and thus, it has limitations inherent to retrospective studies. Given the bivariate model of the meta-analysis, we did not calculate the overall ACC for the differentiation classes. Moreover, our subgroup analysis was limited by the number of studies identified.

Nonetheless, to the best of our knowledge, this is the first meta-analysis to picture the current performance of radiomics for discrimination of pediatric MB, PA, and EP of the posterior fossa, providing cutting-edge conclusions to pilot future models.

The systematic review also included a risk of bias assessment using the NOS. The NOS allowed for the evaluation of the quality of the included studies based on the selection criteria, comparability of the study, and outcome assessment. This assessment ensured that the included studies were reliable and provided robust evidence for the discrimination between these tumors. To further reinforce the transparency of our assessment, we have incorporated a visual representation, shown in Figure 5. This map illustrates the risk of bias in various domains using a “traffic light plot” derived from the robvis library [33], displaying risk judgment in each domain for every study. Green indicates low risk, yellow suggests medium risk (i.e., some concerns), and red highlights high risk. This intuitive graph offers a visual representation of the risk of biased judgments in each domain for a clearer understanding of the methodological evaluation. In conjunction with these graphical representations, our risk-of-bias assessments were conducted leveraging the RoB 2 tool [34].

## 5. Conclusions

Our study revealed that radiomics-based models exhibit promising capabilities in discriminating between the most common pediatric posterior fossa tumor types, with modest variations in performance observed across MB, PA, and EP.

The T1W and T2W sequences, combined with the LR classifier and textural features, yielded the highest ACC for distinguishing MB. Conversely, the T2W sequence with LR and NN classifiers and geometrical/morphological features demonstrated the highest ACC for PA and EP.

Despite the diversity in methodologies and study designs across the included articles, our results consistently highlighted the potential of radiomics as a valuable non-invasive tool for pediatric posterior fossa tumor differentiation. The variability in diagnostic performance across different tumor types underscores the need for tailored approaches in radiomics model development.

Our study encourages further refinement in feature extraction methods, the exploration of hybrid imaging approaches, and the incorporation of larger multicenter datasets to enhance model generalizability. Our insights into the significance of sequence selection, choice of classifiers, and feature extraction guide future investigations.

## Figures and Tables

**Figure 1 cancers-15-05891-f001:**
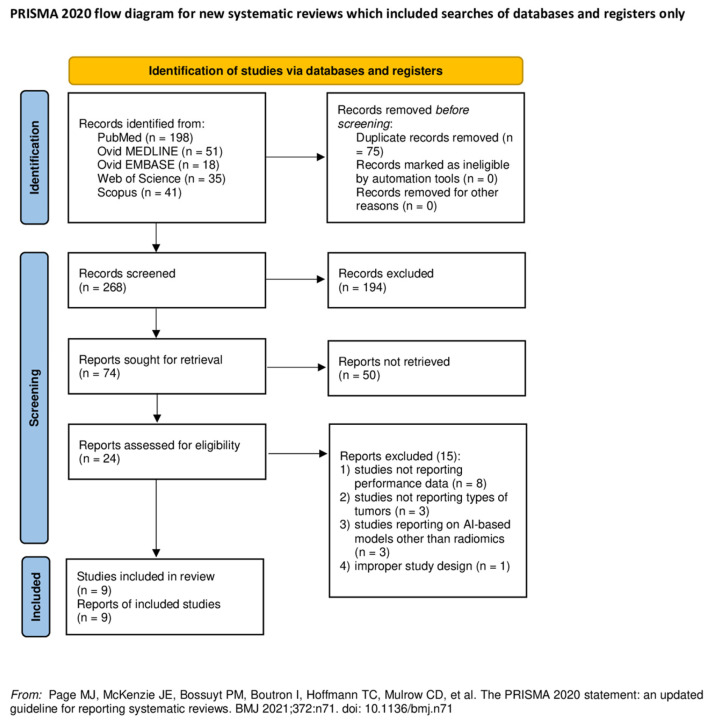
PRISMA flow diagram depicting the literature search process. Abbreviations: PRISMA = Preferred Reporting Items for Systematic Reviews and Meta-Analysis; AI = artificial intelligence [9]. http://www.prisma-statement.org/, accessed on 1 July 2023.

**Figure 2 cancers-15-05891-f002:**
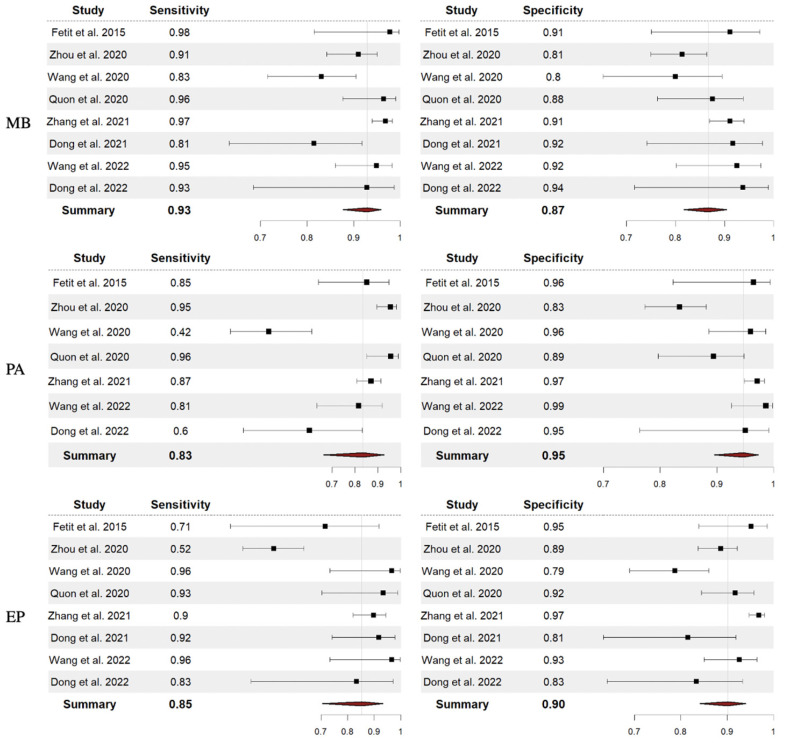
Forest plots with overall SEN and SPE for discrimination between MB, PA, and EP. Abbreviations: SEN = sensitivity; SPE = specificity; MB = medulloblastoma; PA = pilocytic astrocytoma; EP = ependymoma [14,15,16,18,19,20,21,22].

**Figure 3 cancers-15-05891-f003:**
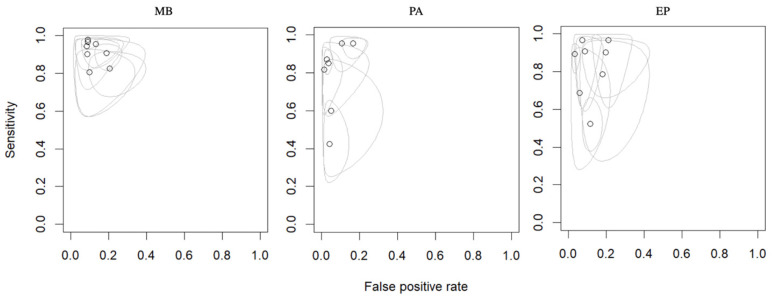
Endpoints of interest with individual confidence regions for differentiation between MB, PA, and EP. Abbreviations: MB = medulloblastoma; PA = pilocytic astrocytoma; EP = ependymoma.

**Figure 4 cancers-15-05891-f004:**
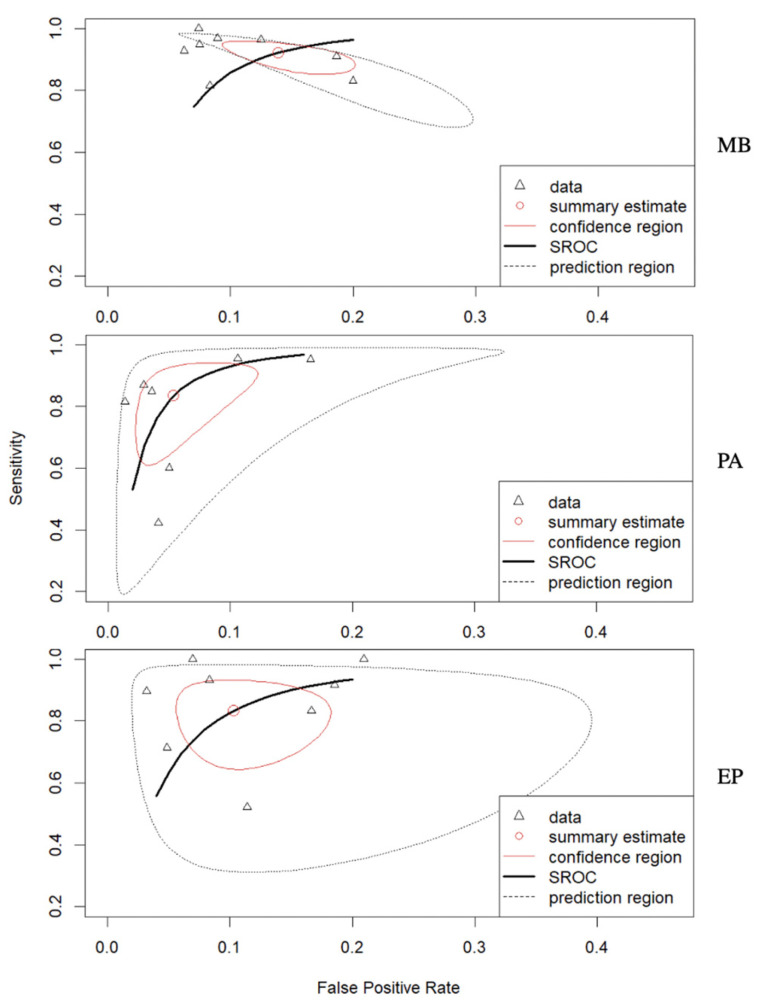
Corresponding SROC curve for the differentiation between MB, PA, and EP. Abbreviations: SROC = summary receiver operating characteristics; MB = medulloblastoma; PA = pilocytic astrocytoma; EP = ependymoma.

**Figure 5 cancers-15-05891-f005:**
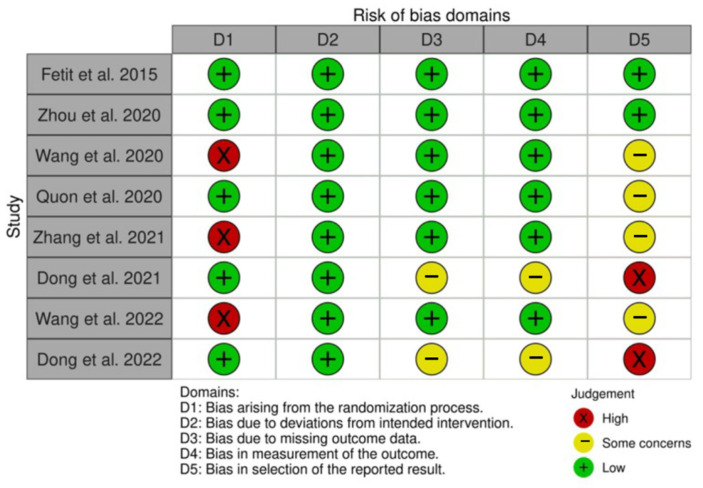
Risk of bias domain map showing the risk of bias assessment of our study. The graph displays the risk of biased judgment in each domain (columns) for each study (rows), leveraging a traffic light color code: green indicates low risk in the given domain for the given study, while yellow and red stems, respectively, from medium risk (i.e., some concerns) and high risk [14,15,16,18,19,20,21,22].

**Table 1 cancers-15-05891-t001:** Summary of the studies.

Author, Journal, Year	Dataset (No.)	MB (No.)	PA (No.)	EP (No.)	Imaging Protocol	Method	Classifiers
Dong, Acad Radiol, 2021 [14]	51	27	-	24	CE-T1W, DWI, ADC	ML	kNN, AdaBoost, RF, SVM
Dong, BJR, 2022 [15]	136	67	37	32	T1W, CE-T1W, T2W, T2-FLAIR, ADC, DWI	ML	SVM
Fetit, NMR Biomed, 2015 [16]	44	21	16	7	T1W, T2W	ML	NB, kNN, CT, SVM, ANN, LR
Grist, Neuroimage clin, 2020 [17]	49	17	22	10	T1W, T2W, T2-FLAIR, CE-T1W, DWI, ADC, DSC	ML	SLNN, AdaBoost, RF, SVM, kNN
Quon, AJNR, 2020 [18]	495	272	135	88	CE-T1W, T2W, ADC	DL	CNN
Wang, Neurochirurgie, 2022 [19]	99	59	27	13	T1W, T2W, ADC	ML	RF
Wang, Zhonghua Yi Xue Za Zhi, 2020 [20]	99	59	27	13	T1W, T2W, ADC	ML	RF
Zhang, Neurosurgery, 2021 [21]	535	278	160	97	CE-T1W, T2W	ML	SVM, LR, kNN, RF, EGB, NN
Zhou, AJNR, 2020 [22]	288	111	107	70	CE-T1W, T2W, ADC	ML	LR

Abbreviations: MB = medulloblastoma; PA = pilocytic astrocytoma; EP = ependymoma; CE-T1W = contrast-enhanced T1-weighted image; DWI = diffusion-weighted imaging; ADC = apparent diffusion coefficient; T1W = T1-weighted; T2W = T2-weighted; T2-FLAIR = T2-weighted fluid-attenuated inversion recovery; DSC = dynamic susceptibility contrast; ML = machine learning; DL = deep learning; kNN = k-nearest neighbor; AdaBoost = adaptive boosting; RF = random forest; SVM = support vector machine; NB = naïve bayes; CT = classification tree; ANN = artificial neural network; LR = logistic regression; SLNN = single-layer neural network; CNN = convolutional neural network; EGB = extreme gradient boosting; NN = neural network.

**Table 2 cancers-15-05891-t002:** Summary of the performance data of the radiomics models.

Author, Journal, Year	SEN	SPE	ACC	PPV	NPV	AUC
MB						
Dong, Acad Radiol, 2021 [14]	0.81	0.92	0.86	0.92	0.81	0.91
Dong, BJR, 2022 [15]	0.93	0.94	0.93	0.93	0.94	1.00
Fetit, NMR Biomed, 2015 [16]	0.98	0.91	0.96	0.91	1.00	0.99
Grist, Neuroimage clin, 2020 [17]	NA	NA	NA	0.86	NA	0.80
Quon, AJNR, 2020 [18]	0.96	0.88	0.92	0.88	0.96	NA
Wang, Neurochirurgie, 2022 [19]	0.95	0.92	0.94	0.95	0.93	NA
Wang, Zhonghua Yi Xue Za Zhi, 2020 [20]	0.83	0.80	0.82	0.86	0.76	NA
Zhang, Neurosurgery, 2021 [21]	0.97	0.91	0.94	0.92	0.96	NA
Zhou, AJNR, 2020 [22]	0.91	0.81	0.85	0.75	0.94	0.94
PA						
Dong, Acad Radiol, 2021 [14]	-	-	-	-	-	-
Dong, BJR, 2022 [15]	0.60	0.95	0.83	0.86	0.83	0.98
Fetit, NMR Biomed, 2015 [16]	0.85	0.96	0.92	0.94	0.90	0.99
Grist, Neuroimage clin, 2020 [17]	NA	NA	NA	0.86	NA	0.80
Quon, AJNR, 2020 [18]	0.96	0.89	0.92	0.86	0.97	NA
Wang, Neurochirurgie, 2022 [19]	0.81	0.99	0.94	0.96	0.93	NA
Wang, Zhonghua Yi Xue Za Zhi, 2020 [20]	0.42	0.96	0.82	0.79	0.82	NA
Zhang, Neurosurgery, 2021 [21]	0.87	0.97	0.94	0.93	0.95	NA
Zhou, AJNR, 2020 [22]	0.95	0.83	0.88	0.77	0.97	0.94
EP						
Dong, Acad Radiol, 2021 [14]	0.92	0.81	0.86	0.81	0.92	0.91
Dong, BJR, 2022 [15]	0.83	0.83	0.83	0.56	0.95	0.94
Fetit, NMR Biomed, 2015 [16]	0.71	0.95	0.92	0.71	0.95	0.99
Grist, Neuroimage clin, 2020 [17]	NA	NA	NA	0.86	NA	0.80
Quon, AJNR, 2020 [18]	0.93	0.92	0.92	0.64	0.99	NA
Wang, Neurochirurgie, 2022 [19]	0.96	0.93	0.94	0.68	1.00	NA
Wang, Zhonghua Yi Xue Za Zhi, 2020 [20]	0.96	0.79	0.82	0.42	1.00	NA
Zhang, Neurosurgery, 2021 [21]	0.90	0.97	0.96	0.86	0.98	NA
Zhou, AJNR, 2020 [22]	0.52	0.89	0.80	0.59	0.85	0.84

Abbreviations: MB = medulloblastoma; PA = pilocytic astrocytoma; EP = ependymoma; SEN = sensitivity; SPE = specificity; ACC = accuracy; PPV = positive predictive value; NPV = negative predictive value; NA = not available.

**Table 3 cancers-15-05891-t003:** Cross-validation analysis, best-performing features, and best-performing imaging modalities of each study.

Author, Journal, Year	Cross-Validation Analysis	Best-Performing Features	Best-Performing Imaging Modalities
Dong, Acad Radiol, 2021 [14]	Yes	Geometrical/Morphological	ADC
Dong, BJR, 2022 [15]	Yes	Voxel intensities	ADC
Fetit, NMR Biomed, 2015 [16]	Yes	Texture	T1W+T2W
Grist, Neuroimage clin, 2020 [17]	Yes	Mean	ADC
Quon, AJNR, 2020 [18]	Yes	Deep features	T2W
Wang, Neurochirurgie, 2022 [19]	No	Voxel intensities	ADC
Wang, Zhonghua Yi Xue Za Zhi, 2020 [20]	No	NA	ADC
Zhang, Neurosurgery, 2021 [21]	No	Geometrical/Morphological	T2W
Zhou, AJNR, 2020 [22]	Yes	NA	NA

Abbreviations: ADC = apparent diffusion coefficient; T1W = T1-weighted; T2W = T2-weighted; NA = not available.

## Data Availability

PubMed: https://pubmed.ncbi.nlm.nih.gov/ accessed on 5 May 2023; Ovid MEDLINE: https://www.wolterskluwer.com/en/solutions/ovid/ovid-medline-901 accessed on 5 May 2023; Ovid EMBASE: https://www.wolterskluwer.com/en/solutions/ovid/embase-903 accessed on 5 May 2023; Web of Science: https://mjl.clarivate.com/home; Scopus: https://www.scopus.com/ accessed on 5 May 2023.

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
