# Peer review of "Radiomics for Differentiation of Pediatric Posterior Fossa Tumors: A Meta-Analysis and Systematic Review of the Literature"

_cancers, 2023, doi:10.3390/cancers15245891_

Round 1
Reviewer 1 Report
Comments and Suggestions for Authors
Title:
Radiomics for differentiation of pediatric posterior fossa tumors: a meta-analysis and systematic review of the literature
Review article:
Interesting and thorough meta-analysis with the objective of evaluating the overall performance of current radiomics-based models for discrimination of pediatric posterior fossa tumors. Based on 24 articles for full-text analysis.
Highlights of the study:
1 Current topic of interest
2 Study design
Weaknesses of the study:
1 Limitations of the study well stated by the authors such as: the meta-analysis was based on retrospective cohort studies
Request for revision:
Accept in present form
Reviewer 2 Report
Comments and Suggestions for Authors
In this study, the authors reviewed and analyzed various radiomics models used to identify three of the most common pediatric posterior fossa tumors. There still are some questions that need to be solved.
1. The major deficiency: selection bias and inclusion of retrospective data. There is a fair amount of statistical corrections that were required for correcting the intrinsic biases from retrospective studies which weakens the overall value/quality of the analysis.
2. The sample sizes of the articles are predominantly small, and the reliability of said articles is somewhat dubious.
3. The radiomics method relies on different types of medical images analyzed with machine learning or deep learning methods. Extensive data sets are required to authenticate the outcome. Because of its inherent bias,it remains questionable whether it can serve as a basis for this paper.
4. Registration is required for meta-analysis.
5. Give the risk of bias domains map.
Comments on the Quality of English LanguageProfessional English editing is required to improve the quality of the paper.
Reviewer 3 Report
Comments and Suggestions for Authors
Please include more discussion regarding the heterogeneity between studies. Are there other tools, e.g., iota-squared (doi.org10.1136/bmj.327.7414.557) that could have been used? It appears that the R package mada was used -- what were the results of the meta-regression?
Is there a more appropriate tool to use besides the Newcastle-Ottawa Scale to determine the quality of the studies, particularly as the authors report that they selected items from the scale because there was no control group? Furthermore, some of the criteria selected clearly are significant only for clinical trials (satisfactory clinical follow up, reported outcomes) -- although there is no agreed-upon checklist for ML studies, tools like QUADAS-2 (doi.org/10.1007/s00134-019-05872-y, doi.org/10.2196/23863) or PROBAST (doi.org/ 10.7326/M18-1377) might be more appropriate for ML meta-analyses.
Please discuss more about subgroup analyses -- e.g., were NN or LR classifiers analyzed separately?
Did any of the studies included perform external validation, or were they only cross validated internally?
Comments on the Quality of English LanguageMultiple minor grammatical errors throughout the paper that should be corrected prior to publication.
Round 2
Reviewer 3 Report
Comments and Suggestions for Authors
The authors have carefully expanded their discussion and addressed all previous concerns.